# Superiority of Mild Interventions against COVID-19 on Public Health and Economic Measures

**DOI:** 10.3390/jpm11080719

**Published:** 2021-07-26

**Authors:** Makoto Niwa, Yasushi Hara, Yusuke Matsuo, Hodaka Narita, Yeongjoo Lim, Shintaro Sengoku, Kota Kodama

**Affiliations:** 1Graduate School of Technology Management, Ritsumeikan University, 2-150, Iwakura-cho, Ibaraki, Osaka 567-8570, Japan; gr0486se@ed.ritsumei.ac.jp; 2Discovery Research Laboratories, Nippon Shinyaku Co., Ltd., Nishinosho-Monguchicho 14, Minami-ku, Kyoto 601-8550, Japan; 3TDB Center for Advanced Empirical Research on Enterprise and Economy, Faculty of Economics, Hitotsubashi University, 2-1 Naka, Kunitachi, Tokyo 186-8603, Japan; yasushi.hara@r.hit-u.ac.jp; 4Retty, Inc., Sumitomo Fudosan Azabujuban Building 3F 1-4-1 Mita, Minato-ku, Tokyo 108-0073, Japan; yusuke-matsuo@retty.me (Y.M.); hodaka-narita@retty.me (H.N.); 5Faculty of Business Administration, Ritsumeikan University, 2-150, Iwakura-cho, Ibaraki, Osaka 567-8570, Japan; lim40@fc.ritsumei.ac.jp; 6Life Style by Design Research Unit, Institute for Future Initiatives, University of Tokyo, 7-3-1 Hongo Bunkyo-ku, Tokyo 113-0033, Japan; ssengoku@ifi.u-tokyo.ac.jp; 7Center for Research and Education on Drug Discovery, The Graduate School of Pharmaceutical Sciences in Hokkaido University, Sapporo 060-0812, Japan

**Keywords:** COVID-19, novel infectious diseases, public health and social measures, non-pharmaceutical intervention, system dynamics, people flow analysis, electronic word-of-mouth

## Abstract

(1) Background: During the global spread of COVID-19, Japan has been among the top countries to maintain a relatively low number of infections, despite implementing limited institutional interventions and its high population density. This study investigated how limited intervention policies have affected public health and economic conditions in the COVID-19 context and aimed to gain insight into the effective and sustainable measures against new infectious diseases in densely inhabited areas. (2) Methods: A system dynamics approach was employed. Qualitative causal loop analysis and stock and quantitative flow model analysis were performed, using a Tokyo Metropolitan area dataset. (3) Results: A causal loop analysis suggested that there were risks in prematurely terminating such interventions. Based on this result and the subsequent quantitative modeling, we found that the short-term effectiveness of a short-term pre-emptive stay-at-home request caused a resurgence in the number of positive cases, whereas an additional request provided a limited negative add-on effect for economic measures (e.g., number of electronic word-of-mouth communications and restaurant visits). (4) Conclusions: These findings suggest the superiority of a mild and continuous intervention as a long-term countermeasure under epidemic pressures when compared with strong intermittent interventions.

## 1. Introduction

In the international context, social responses to COVID-19 have comprised various non-pharmaceutical interventions (NPIs) [1,2], including lockdowns. Although these measures may help prevent the spread of infection during the acute disease phase, many strong NPIs have resulted in obvious negative effects [3]. This highlights the need to focus on NPI optimization, especially to achieve sustainable social responses against new infectious diseases.

In this study, we focused on Japan, which is an Organization for Economic Co-operation and Development (OECD) country that has implemented various tiered countermeasures designed to minimize the economic impacts of COVID-19 [4]. Despite the early outbreak of COVID-19 across many densely populated urban areas [5], Japan has kept its infection numbers low, particularly when compared with the numbers in other early breaking countries (Figure 1). As for specifics, Japanese NPIs have discouraged visitation to crowded locations [6] through so-called “mild lockdowns”, which refer to non-coercive stay-at-home requests [7]. Because of the low numbers seen across Japan, it is highly important to analyze the dynamics of these mild lockdown conditions. In this context, the first hypothesis considered in this research was that a mild and continuous lockdown has superiority over a strong and intermittent lockdown. Such information would be of value in implementing countermeasures with minimized negative economic impact on restaurant industry prior to the subsequent sub-acute/chronic phase of COVID-19 control (e.g., mild lockdowns and/or new-normal lifestyles) when increased disease resistance may be achieved through vaccination.

Previous studies have extensively investigated the negative effects of COVID-19 on different areas of business, especially in the service industry. For example, Yang et al.’s [8] investigation of the restaurant industry confirmed that stay-at-home orders led to lower demand; here, voucher programs were suggested as appropriate relief strategies.

As another example in the service industry, Anguera-Torrell et al. [9] examined the effects of COVID-19, economic stimulus, and government-sponsored loans on stock prices in the hotel industry. COVID-19 presented negative effects, while loans showed positive effects. In particular, the restaurant industry should also develop contingency plans in which transactions are targeted at takeout and delivery services, thereby minimizing the need for human contact [8].

In this regard, contactless and/or digital payment technologies can produce reinforcing effects. Another contactless example pertaining to sales and advertising is communication via electronic word-of-mouth (eWOM), which facilitates social distancing. Previous research has shown that eWOM communication positively impacts sales [10]. In this context, the second hypothesis to be addressed in this research is that increased sales may be achieved through a concomitant rise in eWOM popularity even during COVID-19. This makes it especially important to explore the dynamics of eWOM communication in the pandemic context.

Based on the information above, this study investigated both the effect of NPI strategy on infection and the restaurant business in the Tokyo Metropolitan area. This analysis would contribute to exploring the way to control infections with minimized negative economic impact on restaurant industry in densely populated areas.

From a technical perspective, we aimed to handle the potential of different responses to public health and economic issues found in the complex metropolitan dataset by implementing structured models. In this context, we investigated how several types of NPIs affected public health in general (based on the number of patients) and specifically within the restaurant industry (using metrics such as the number of customer visits and eWOM communications).

As a research strategy, we employed system dynamics, which is a powerful tool for investigating “what-if” scenarios in complex situations. System dynamics is a method developed by J.W. Forrester for dealing with questions about the dynamic tendencies of complex systems. Although this approach utilizes quantitative models, it should rather be regarded as an extension of the case study method, as it utilizes mental observation or experience (mental database) rather than actual numerical data [11]. Using this, this study aimed to asymptotically approach the previously stated hypotheses that (1) mild and continuous lockdown has superiority over strong and intermittent lockdown, and (2) increased sales may be achieved through a concomitant rise in eWOM popularity even during COVID-19.

## 2. Materials and Methods

### 2.1. Data Sources

In this study, demographic data were obtained from a portal site containing official statistics relevant to Japan (e-Stat) [12]. The data were collected by the Ministry of Internal Affairs and Communications. We obtained the baseline frequency of dining out and statistical data pertaining to restaurant business from the Foodservice Industry Research Institute [13].

Next, eWOM and customer visit metrics were gathered from a database compiled by Retty Inc., which runs an integrated web-based eWOM and restaurant reservation service. In this study, eWOM was electronically reported by a qualified reviewer, and customer visit metrics were also collected electronically.

People flow metrics were obtained from Agoop, which is a big data company that collects location information from smartphones [14].

Finally, data related to consumer sentiments were obtained through a monthly survey conducted by the TDB Center for Advanced Empirical Research on Enterprise and Economy (TDB-CAREE), Hitotsubashi University, Tokyo, Japan [15].

### 2.2. System Dynamics Model Linking Disease Spread, Non-Pharmaceutical Countermeasures, and Economic Effects

Initially, we conducted a causal loop analysis to investigate the semi-quantitative aspects (i.e., ascending or descending) in consideration of important identified factors. This analysis focused on detecting possible causal relationships between Japanese NPIs and items of interest (infection, people flow, restaurants visit, and eWOM) based on preceding studies. Logical relationships were also used to identify causality, based on the concept of “mind model” constituting the basis of system dynamics [11]. Details of information used in constructing a causal loop are described in the results section (Section 3.2, Section 3.3, Section 3.4, Section 3.5 and Section 3.6).

Next, we employed a quantitative stock-flow model. In the stock-flow model, we constructed the (1) disease spreading part, (2) NPI effects on people flow part, and (3) restaurant visits and eWOM part. This quantitative modeling was based on epidemic data obtained from the Tokyo Metropolitan area, specifically on Japanese NPI practices from March to September 2020. Meanwhile, the data used to explore eWOM dynamics were collected from 2019 to 2020. Appendix A shows the basis for our quantitative model parametrization [16,17,18].

In the disease-transmission part, transmission efficiency was described as the number of cases generated by one carrier and set to basic reproduction numbers under no affecting factors. Meanwhile, affecting factors were composed of people flow and behavior components. The people flow component consisted of people flow in crowded places and a scaling factor that related basic reproduction numbers to real conditions. People flow that was responsible for transmission was represented by maximum people flow (persons per hour) on Wednesdays (middle of the week). The behavior component was composed of several parameters, including distancing and protective behavior, epidemic consciousness, fear of infection, and protective behavior completeness; here, the effects were represented by relative values ranging from 0 to 1. As non-pharmaceutical countermeasures, stay-at-home requests and behavior guidance were considered modulators of people flow and behavior, respectively. Remote-work practices were also thought to affect people flow.

The economic effect part described economic effects on restaurant industry, including interaction with eWOM. Here, the concept was that customer visits were affected by stay-at-home requests, epidemic consciousness, and positive eWOM. Restaurants with eWOM were assumed to be less susceptible to people flow and reduced people flow by remote work, because visiting restaurants with reputation is considered to be an activity with strong intention. We also assumed that restaurants with positive eWOM had better sales based on preceding study [10].

Both the causal loop and quantitative stock and flow models were built by using Vensim PLE version 8.0.9 (Ventana Systems Inc.), while data summarization was accomplished by using Microsoft Excel 365 (Microsoft Corporation).

### 2.3. Reality Checking and Refinement of the Quantitative Model and Further Simulation

To check the reality of the model structure, we used the concept of system dynamics, which is an extension of the case study approach [11]. Using the constructed model, time-course profiles between the real data and model output were compared (Appendix A). We first confirmed that the quantitative model had sufficiently good fit and that the approach was adequate. However, there were two notable limitations. First, the model tended to underestimate patients, as neither within-family transmission nor the cluster effects were modeled. However, the ability to describe historical disease spreading events implies that the model structure was useful for determining contributions and describing the dynamics of random transmission in crowded cities. Second, we did not adequately model the sharp decrease in customer visits found during April 2020. This may have been the result of a shock effect in which the population was facing a pandemic for the first time. As such, additional research should target the factors influencing these dynamics.

### 2.4. Refinement of the Quantitative Model and Constructing the Hypothetical Model and Simulation

After constructing the quantitative model, we simulated the NPI effects based on different strategies and strengths, and we observed the dynamics of both eWOM communications and customer visits to restaurants.

Although this simulation required resolving the effect of each NPI (observed effect was possibly a combination of several NPIs’ effects), some of the NPI effects could not be reasonably resolved because of the lack of sufficient time-course data with various conditions. The simulation was performed under the assumptive effect sizes of each NPI, while the effect sizes of school closures and stay-at-home requests were 20% and 10%, respectively. These NPI factors similarly affected people flow, customer visits, and eWOM communications. An NPI focused on suspending night services at restaurants was set to decrease only customer visits and eWOM communications (10% each).

This simulation also required model consistency over a time period of interest. To describe long-term dynamics on people’s behavior, the hypothetical factor of epidemic consciousness was introduced. Using this, the quantitative model was re-built as a hypothetical model, still maintaining the feature of a “mind model” in the system dynamics context. Short-term epidemic consciousness was set to decrease people flow by 10%, while mid-term pandemic consciousness was set to decrease customer visits and eWOM by 20% each, and long-term pandemic consciousness was set to decrease customer visits and eWOM by 30% and 20%, respectively. Model parameters related to NPI scenarios are provided as Appendix A.

## 3. Results

### 3.1. Observation on NPIs and Their Effects on People Flow and Restaurant Industry

People flow refers to intercommunity human contact and can thus play a significant role in disease transmission, particularly in crowded areas. This is especially pertinent in Japan, which contains several cities with high population densities [5]. As such, we investigated the dynamics of people flow using data provided by Agoop, which is a big data company that collects location information related to people flow through smartphones [14].

For example, we examined the time course of people flow at Shinjuku Station in Tokyo, which is known for facilitating the largest number of commuters in Japan (as many as 250,000). People flow first decreased in March 2020, just after the ministry directed school closures. There was a drastic decrease in April 2020, when the stay-at-home request was issued. Figure 2 shows changes in the number of people at 18:00 on weekdays, which is considered the peak both around and within Shinjuku Station; this represents people flow during the typical commuting time in the Tokyo Metropolitan area. Even after the stay-at-home request was lifted, people flow was found to be at considerably lower levels when compared with pre-pandemic numbers. This shift to lower levels may indicate a general transition to a “new-normal” lifestyle (e.g., remote work or behavioral changes in which people remain home or avoid visiting crowded locations due to long-term concerns about COVID-19).

In this context, we constructed quantitative models to test the hypothesis that people flow in crowded places can affect disease transmission. This is described in more detail later in the manuscript.

To understand how NPIs affect the restaurant industry, we analyzed data on actual customer visits and eWOM communications from Retty Inc., which runs an integrated web-based eWOM and restaurant reservation service [19]. Both monthly metrics were taken from the year 2020 and normalized on the basis of the previous year. In short, results suggested that customer visits and eWOM (together with people flow) decreased because of COVID-19 in both March and April 2020, with further possible effects from the stay-at-home request issued in April 2020 (Figure 2). All three metrics were similarly affected by these events. Using monthly data from March to September 2020, the correlation coefficient between people flow and customer visits was 0.916, while that between people flow and eWOM was 0.879.

Further, the move in eWOM communications was slower than the shift in customer visits during the pandemic phase from March 2020 to May 2020 (which includes the stay-at-home request). This suggests that, although eWOM follows customer visits, there is a delay in outcomes during crisis-driven circumstantial changes. Another hypothesis is that eWOM is resistant to pandemic crises because of the remote nature of the information transmission.

In general, we found that eWOM communications and customer visits were similarly affected by initial information and interventions related to COVID-19; indeed, full recovery was not observed for six months (October). This suggests that the pandemic has had long-term effects on public consciousness.

In sum, eWOM did not appear to affect customer visits in Tokyo based on the abovementioned time course dynamics. Rather, there was a tendency in which eWOM followed customer visits during the first pandemic wave. Although previous reports have shown that eWOM affects sales [10], most such findings are based on cross-sectional observations without clarifying specific conditional dynamics. This suggests the need for additional research once time-course data have been collected under various conditions.

From another point of view, customer visits and eWOM communications did not recover to the level before the pandemic (the year before), suggesting some long-term effects on human behavior.

### 3.2. Causal Loop Diagram: Dynamics of Infection

The causal loop diagram of the dynamics of infection is shown in Figure 3B. The transmission of the virus has resulted in what is known as the COVID-19 pandemic; in this regard, it is certain that SARS-CoV-2 spreads during close interpersonal contact [20]. The dynamics of COVID-19 transmission have successfully been described through the use of models that show that infections are spread during such exposure, especially among asymptomatic carriers and susceptible populations [21,22].

The information that transmission efficiency may be affected by travel [21] and population density [22,23,24,25] also suggests that human-to-human contact is important in the transmission of viruses.

In sum, the current literature shows that human-to-human contact is a determinant of transmission. As such, the course of infection (including the role of asymptomatic carriers) was featured in the causal loop.

### 3.3. Causal Loop Diagram: Reviewing the Basis of Japanese NPI

Causal loop diagrams of Japanese medical systems and NPIs are shown in Figure 3C,D, respectively. In constructing these diagrams, the whole picture of Japanese NPIs in relation to medical systems was reviewed.

Regarding those implemented in 2020, Japanese NPIs can be categorized on the basis of requirement for behavior, including mild tiered interventions (raising sanitation awareness, physical distancing, encouraging remote work, and staying home), focused NPIs in high-risk settings (suspending night services for bars and restaurants), and strong NPIs in more severe cases (strongly asking people to remain home and closing businesses temporarily, as per administrative order) [26].

Milder NPIs were implemented during the early phase of the domestic outbreak in February 2020, at which time behavioral requests were issued, such as remembering to wash hands and covering the mouth when coughing [4]. More stringent requests were introduced in March 2020, when individuals were advised to avoid crowded locations [6].

Disease spread in the next phase resulted in a shortage of hospital beds. As of April 2020, only 12,500 beds were available for those with novel infectious diseases in Japan, accommodating approximately 10,000 patients at the time [22]. The surge of COVID-19 infections thus impelled the Japanese government to declare a state of emergency, at which time hospitals were still making the effort to increase capacities [27]. There were also concerns about the possible need to accept additional patients during subsequent epidemic phases, thereby putting health systems in further danger.

Subsequently, stronger NPIs were implemented on the basis of the declared state of emergency, from a legal perspective of facing disease severity, uncontrollability, and the potential of overwhelming hospitals. In addition, a second state of emergency was declared in January 2021 as Japan entered the next wave of the pandemic and faced a shortage of hospital beds.

Nevertheless, it is generally understood that NPIs may negatively affect the economy because of restricted mobility, especially during lockdowns [28]. In Japan’s case, economic countermeasures were targeted at the travel industry, which was expected to benefit from anti-epidemic prioritization in the NPI context in July 2020, when the number of patients in Tokyo (the most densely inhabited area) indicated the existence of a second viral wave [26]. This is evidence of how the economic sector may pressurize the government to reduce the intensity of NPIs, which may negatively impact the economy, and to require economic measures in times of crises.

From the above, we considered that hospital-bed shortages were key factors for inducing strong NPIs. On the other hand, we believed that concerns over economic stagnation would serve as negative feedback to stricter NPIs. As a part of its public health policy, Japan’s virus testing operations are focused on potential and identified clusters rather than mass testing; this is also a component of an active epidemiological survey [29]. In this regard, it would be adequate to consider that identifying virus carriers by either mass-oriented or focused approaches will help quarantine. All of these aspects were featured in the causal loop diagrams.

### 3.4. Causal Loop Diagram: Behavioral Effect on Transmission and Triggering Behavioral Change

In constructing the behavior part of Figure 3D, the following items were considered. Among the various behavioral measures taken to avoid viral transmission, a systematic review has suggested the particular efficacy of physical distancing and the usage of face masks [20]. In Japan’s case, previous studies have investigated behavioral changes during the very early phase (February 2020), thus highlighting the positive initial response triggered by mass-media reports of substantial infections incurred on a cruise ship, in which case the ministry’s order to close schools was well-observed [30,31]. Further, an online survey conducted by the LINE Corporation and Ministry of Health and Welfare revealed behavioral changes in which individuals were more willing to avoid crowded locations as soon as Japan declared a state of emergency (April 2020) [32]. From these episodes, it would be adequate to consider that behavioral changes are induced via mass-media reports on disease threats and the need for strong NPIs. Thus, the above items were included in the causal loop diagram.

### 3.5. Causal Loop Diagram: People Flow

In constructing the people-flow part of Figure 3D, people flow was considered to refer to intercommunity human contact and was thought to play a significant role in disease transmission as described before.

### 3.6. Causal Loop Diagram: Customer Visits and eWOM

A causal loop diagram related to eWOM is shown in Figure 3E. Reduction in eWOM mass can lead to a reduction in customer visits considering the positive relationship between eWOM and sales [10], but the supporting information for this hypothesis has not been obtained till date (described in broken line). Because business closure, as a part of the economic impact, is related to the reduction in strong NPIs, this linkage is described in solid lines.

### 3.7. Constructing an Integrated Causal Loop Diagram

Based on the information above, we constructed a causal loop diagram depicting the general situation for COVID-19 transmission in the context of Japanese NPIs, including the drawbacks (Figure 3A). Several feedback loops are present in the diagram. As for disease infection, COVID-19 transmission showed a reinforcing nature during the disease spreading phase (Figure 3B, upper); theoretically, herd immunity also appeared to decrease the transmission rate (Figure 3B, lower). On the other hand, the medical collapse loop illustrates the inadequacy of utilizing herd immunity as a political strategy (i.e., allowing infections; Figure 3C, lower), specifically showing that such practices are likely to result in uncontrollable transmission rates and insufficient medical treatments; in turn, this leads to a considerable increase in the death rate. An active epidemiological survey, which is a focused investigation on the circumstances of infection related to patients, found that including virus testing of people in contact with patients [33] would partially decrease the transmission rate (Figure 3C, upper). As shown in Figure 3D, appropriate balancing dynamics and the strength of NPIs both decrease when the infection rate decreases. In contrast, the lower portion of Figure 3E indicates a negative feedback for strong NPIs based on economic stagnation, which can result in the premature termination of NPIs from a public health perspective. Finally, the upper portion of Figure 3E shows the effects of eWOM. Previous research has clarified that increased eWOM is related to better sales [10]. In this study, we hypothesized there would be a short-term increase and decrease of eWOM mass in relation to sales.

While the nature of the 2020 pandemic has highlighted the need to prevent hospitals from becoming overwhelmed, there is still some reticence towards stronger NPIs, which can lead to economic stagnation. By contrast, delayed NPI implementation may worsen the COVID-19 mortality rate [23]. Practically speaking, a large number of hospital beds will be occupied for longer durations than normal, which will put serious pressure on hospital management. These conditions further emphasize the need to adequately time the implementation of appropriate NPIs. In this study, we investigated the issue by conducting a quantitative simulation, which is described in the following sections.

### 3.8. Quantitatively Modeling the Disease Transmission Dynamics

According to our model, disease transmission was accomplished via interpersonal contact, which was affected by both people flow and protective behavior. Here, we assumed that people flow was affected by pandemic consciousness, stay-at-home requests, and new-normal lifestyle effects. The structure of the final quantitative stock-flow model, including all items added in the processes appearing in Section 2.2, Section 2.3 and Section 2.4, is shown in Figure 4. The right side of Figure 4 expresses the disease-transmission part, which is connected to the people flow and behavior part on the left.

Short-term pandemic consciousness was a hypothetical psychological factor introduced to explain community resistance against disease based on behavior change. In this study, it was introduced to explain variable transmission efficiency over time, according to increases or decreases in reported patient numbers, as shown in previous research [22,34]. We hypothesized that media reports of increased cases would increase the level of pandemic consciousness, thus catalyzing risk-evasive behaviors (e.g., not going out or wearing face masks and washing hands more frequently). Behavioral actions were parameterized as protective behavior, and then calibrated along the time course of real patient numbers; this effect was assumed to be reversible. The probability to engage in protective behavior was set to 0.6 for the pandemic condition. This was derived on the basis of the results of a survey conducted by TDB-CAREE [15], which reported that about 60% of respondents believed more stringent measures were necessary in Tokyo as of June 2020. This subpopulation was therefore considered more likely to engage in protective behavior. Meanwhile, new-normal lifestyle effects included the tendency to engage in remote work. This was handled as non-reversible based on the recognition that barriers to remote work included cyber security issues and employment rules.

As mentioned earlier, mass viral screening was not implemented in Japan, where virus testing was instead limited to symptomatic patients. During the early stages of the outbreak, typical symptoms were considered fever, fatigue, and/or shortness of breath. Our model was constructed in accordance with these conditions.

### 3.9. Integrating a Quantitative Systems Model across Disease Transmission, People Flow, and the Restaurant Industry

Our quantitative stock and flow model (Figure 4) consisted of three components, including a disease transmission model, people flow and behavior model, and effect on the restaurant industry model. First, the disease transmission model part was built to explain that virus carriers would transmit the virus to susceptible persons. We modeled symptomatic and non-symptomatic infections, using data showing confirmed positive cases based on Japanese practices (i.e., virus testing was fundamentally limited to confirmative testing for symptomatic patients). Within the model, infections were considered to be dependent on the basic reproduction number [35], interpersonal contact, temperature [23], and the proportion of susceptible persons (i.e., non-immunized). On the other hand, we did not include virus mutations, possible vaccinations, or mass virus screening.

Second, the people flow and behavior model part was built to explain that interpersonal contact related to disease transmission was affected by maximum people flow (i.e., locations for disease transmission) and personal behavior (personal protective measures). Within the model, both elements were affected by psychological factors and short-term pandemic consciousness. Meanwhile, people flow was further dependent on extrinsic factors, such as stay-at-home requests and new-normal lifestyle effects, while behavior was further affected by the distancing effect, behavior guidance, and the thoroughness of protective behavior.

Third, the restaurant industry model part was built to explain that customer visits to high-grade restaurants were dependent on the intention to dine out, but not necessarily dependent on people flow. As the interactions between customer visits and eWOM were unclear, they were modeled as being independently affected by similar factors, such as stay-at-home requests, focused intervention effects, mid-term pandemic consciousness, long-term pandemic consciousness, and the psychological effect of school closures. Here, mid-term pandemic consciousness refers to a continuous mindset spanning months, particularly concerning the idea that individuals should voluntarily refrain from going out due to the risk of spreading disease. Next, long-term pandemic consciousness refers to a similar mindset that remains continuous for at least six months. This idea was introduced on the basis of the observation that customer visits and eWOM appeared to steadily react in contrast to fluctuating people flow. Finally, the psychological effect of school closures refers to both an initial recognition of the pandemic based on ministry-directed school closures [30,31] and a continued hypothetical psychological effect in which individuals avoid dining out as long as schools and other important educational facilities remain closed.

Though some part of the model structure was hypothetical, the quantitative aspects were calibrated based on real metrics, thus enabling useful simulations. In this regard, real conditions were investigated through real data, which were also used as a basis of comparison for the number of observed patients (confirmed positive virus infection cases), people flow (obtained via smartphone location information), number of visits, and eWOM communication under realistic conditions.

### 3.10. NPI Pattern Simulation

We tested the effects of the four following interventional conditions: (A) realistic conditions, in which one stay-at-home request lasting 1.5 months was issued in April 2020 (first pandemic wave). However, countermeasures against the second wave were limited to appeals for protective behavior and remote work; (B) hypothetical stay-at-home request lasting 1 month was issued in July 2020, specifically as a countermeasure against the second pandemic wave; (C) hypothetical stay-at-home request lasting 1 month was issued in June, specifically as a pre-emptive countermeasure against a second pandemic wave; and (D) an exhaustive intervention scenario, in which a first stay-at-home request lasting 2 months was issued in March 2020, specifically as a pre-emptive countermeasure against a first pandemic wave, with a second stay-at-home request lasting 2 months being issued in July 2020.

The number of patients (evaluated as confirmed positives) differed between scenarios (Figure 5A), with stay-at-home requests (especially when pre-emptively issued) lowering the number of patients. On the other hand, the more exhaustive intervention with stay-at-home request tended to result in more negative economic effects (Figure 5B), as represented by reduced customer visits. The outcome that was based on the direct effect size of the stay-at-home request was parameterized as a 10% decrease in customer visits and eWOM communication. More specifically, this parameterization was based on the consideration that stay-at-home requests without closures (as actually directed in January 2021) do not constitute strong interventions. For reference, a previous study found a small add-on effect related to the lockdown condition [1].

There were two important findings. First, scenario (B) (pre-emptive stay-at-home request to counteract the second pandemic wave) effectively controlled the pandemic in the short-term context, with only small negative impacts to restaurant businesses; however, prematurely lifting the request would cause an explosive growth in the number of infections. Second, based on the current effect size of the employed factors, the economic effects of an additional lockdown were small, but the anti-pandemic effects were large. These findings indicate that a mild lockdown of substantial duration is an effective way to curtail the effects of the virus.

## 4. Discussion

From the initial observation of the data obtained, people flow, customer visit to restaurants, and the number of eWOM showed a similar time course during and after the stay-at-home request phase. No supporting information showing the effect of eWOM on customer visit was observed under this study. On the other hand, customer visits and eWOM communications did not recover to normal condition, suggesting some long-term effects of NPIs on human behavior.

Our causal loop analysis suggested that the economic sector may pressurize the government to reduce NPI measures in times of crises. This could urge societies to avoid or prematurely terminate strong NPIs, which are often necessary during viral pandemics. To avoid this, there must be ways to increase resilience among businesses that are vulnerable to new infectious diseases. This is particularly important for those operating in the restaurant, hotel, and travel industries. The appropriate use of mass media is also important, as news coverage is likely to influence behavioral change. Compared with the rates seen in other OECD countries, the relatively low number of COVID-19 patients found in Japan may have partially been the result of increased pandemic consciousness stemming from news coverage showing mass infections on cruise ships, as well as the implementation of various administrative orders, including school closures and stay-at-home requests [30,31]. Regarding regional differences, it should be noted that there is room for research on biological ethnic differences, such as pre-existing immunity [36]; however, we avoided the influence of factors that lead to severe outcomes by focusing on the number of positives.

Next, our quantitative model showed that pre-emptive and strong short-term interventions were efficient ways of controlling the spread of disease over short periods of time with minimum economic effects. However, these policies failed to control the spread of disease in the long-term context. The short-term efficiency of pre-emptive shorter lockdowns can be explained on the basis of the results of a preceding study in which a cross-regional analysis [20] revealed that earlier lockdowns were more effective, while one-week delays in lockdown orders resulted in considerable disease spreading. In other words, while pre-emptive measures are effective, early lockdown terminations will still reboot the viral spread.

In sum, our simulations suggested that additional stay-at-home requests without school closures were effective, with only small negative effects. This was based on an assumption we built into the model: that is, initial fear and pandemic awareness triggered by mass media coverage and the ministry’s order to close schools created an environment in which the public thoroughly followed recommended behaviors, such as engaging in personal protective measures. These results were also based on the recognition that it takes at least six months for customer visits and eWOM communications to recover from pandemic awareness. The analysis was further based on the assumption that only small negative impacts would be incurred through additional NPIs. Here, the efficacy of these additional interventions (typically implemented within six months of the initial intervention) constitutes an important practical finding. Combined with the usefulness of assessing the performance of various NPIs both alone and combined [1,2], as well as the flexibility of their operation [28], this finding should therefore facilitate the reduction of economic losses during NPIs.

As for practical insights, we found that the timeframes used for NPIs were highly important in the context of controlling new infectious diseases. Specifically, we recommend against the early termination of lockdowns, which is likely to result in the need for repeated implementation and lifting; this is not sensible. On the other hand, we recommend the continuation of mild sustained interventions, which are more likely to achieve long-term consistency in both public and economic health.

This study also had some limitations. One, the model did not explain either within-family transmission or the cluster effects and tended to underestimate patients as shown in Figure 5A. Reliable effect sizes were not available for each NPI, in which case additional research should conduct time-course analyses under different conditions. Two, the studied time-course lasted from March 2020 to September 2020 in order to concentrate on the short-term effects of the pandemic and initial NPI. In this context, it is unlikely that we captured the long-term effects. Future studies should therefore focus on administrative activities that can support vulnerable industries. Nevertheless, our simulations employed a deductive model with hypothetical features that were supported through quantitative calibration, and we were thus able to clarify complex societal systems and possible linkages between factors. This approach would also be useful in research targeted at the dynamics of other OECD countries, thereby pointing out areas of industrial similarity and/or social differences.

## Figures and Tables

**Figure 1 jpm-11-00719-f001:**
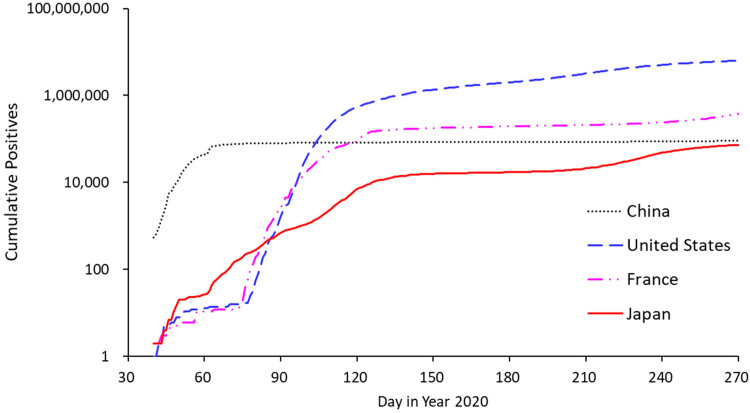
Number of cumulative positive COVID-19 cases in representative early breaking OECD countries (United States, France, and Japan) and positive cases from the epicenter in China. Cases in the epicenter were reduced through strong interventions, while the proportion reached 10% in the United States (highest national proportion). The increase was more gradual in Japan.

**Figure 2 jpm-11-00719-f002:**
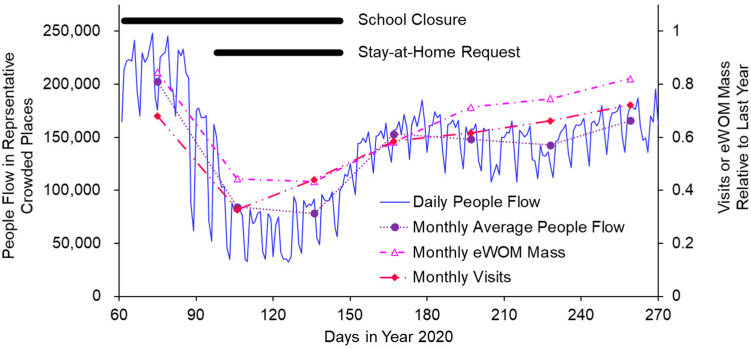
People flow represented by the number present at 18:00 on weekdays, when there is typically a peak around and within Shinjuku Station (500-m radius from the station center) in the Tokyo Metropolitan area (left *y*-axis); customer visits and electronic word-of-mouth (eWOM) mass (right *y*-axis), recorded in 2020. Data were collected via smartphone location information, integrated web-based eWOM, and restaurant reservation services.

**Figure 3 jpm-11-00719-f003:**
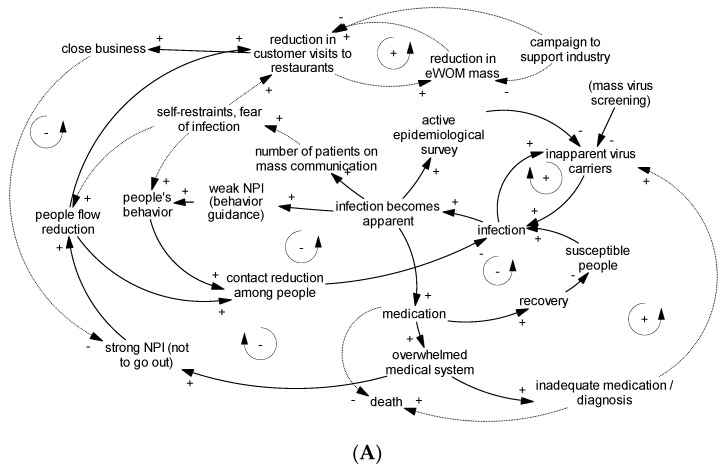
Causal loop diagram for COVID-19 transmission reflecting Japanese non-pharmaceutical interventions (NPIs). Solid arrows indicate relationships supported by theories or previous studies, while broken lines indicate hypothetical relationships, both reflecting the mind model in the system dynamics context. Restaurants were deemed representative businesses. Each panel shows (**A**) the whole picture, (**B**) reinforcing the infection spreading loop and balancing the herd immunity loop, (**C**) balancing the active epidemiological survey loop and reinforcing the medical collapse loop, (**D**) balancing the people’s behavior loop and strong NPI loop, and (**E**) reinforcing the electronic word-of-mouth (eWOM) loop and balancing the negative effects on strong NPI loop.

**Figure 4 jpm-11-00719-f004:**
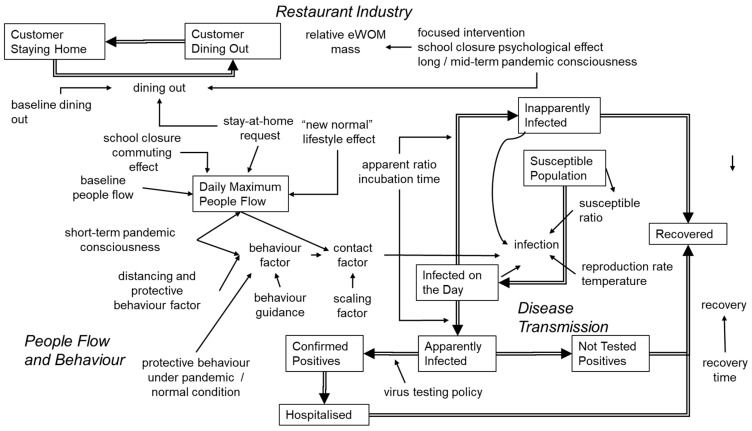
Integrated quantitative systems model across disease transmission, people flow, and the restaurant industry.

**Figure 5 jpm-11-00719-f005:**
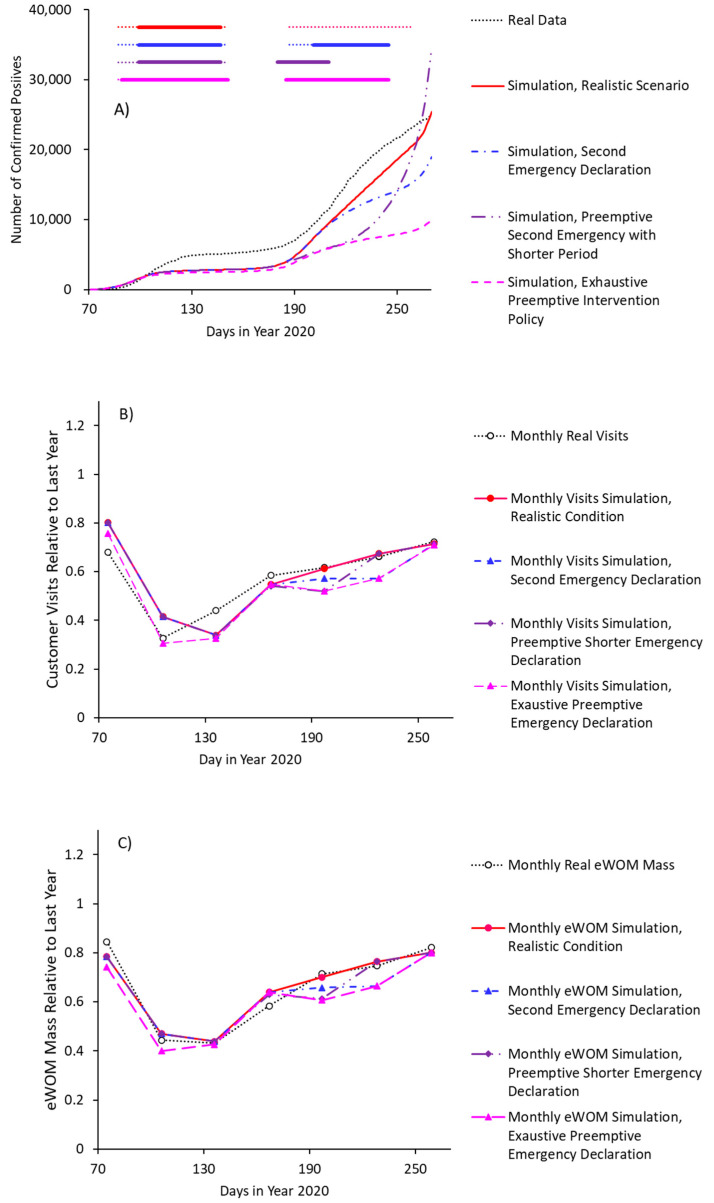
(**A**) Simulation outcomes of confirmed positives, (**B**) customer visits, and (**C**) electronic word-of-mouth (eWOM) mass. Colored bars on the upper side of the panel show duration of stay-at-home request (thick bars) and epidemic consciousness raised by information from 1st- and 2nd-wave outbreaks (thin dotted bars) in each scenario (3rd-wave outbreak was not taken into consideration).

## Data Availability

Data used in the analysis reported are available as Appendix A.

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
