# Peer review of "Superiority of Mild Interventions against COVID-19 on Public Health and Economic Measures"

_jpm, 2021, doi:10.3390/jpm11080719_

Round 1

Reviewer 1 Report

Peer review of article titled Superiority of mild interventions against COVID-19 on public health and economic measures.

  1. Introduction

I found this section to be weak because it did not lay our clearly what the research questions were to be addressed in the study.

Specific areas that were unclear:

Line 55-59: this long sentence contains far too many ideas.

Line 65-80: the mix of ideas are difficult to understand what the key point is.

Line 83: what is sustainable infection control? Definitions are needed throughout the paper.

Line 89-98: this information should be in Materials and Methods.

  1. Materials and methods

Line 116-122: How were the causal loop diagrams developed? What was the exact process? No idea what bottom-up means. How was data summarized? How was the stock and flow diagram created? What was the data analysis process?

Line 137: what economic effect model?

Line 139: assumed? Could you revise the paper and clearly articulate hypothesis examined?

Line 145: real and simulated data? Not idea what this means here. The paper needs a clear section on how each data set was identified and analyzed.

Results

Line 170: I don’t think there is any debate anymore about aerosol transmission!

This first paragraph does not clearly explain results.

I had a very hard time reading the results section and often myself questioning how various paragraphs fit with what I understood to be the study design – comparing “dynamics of infection, NPIs, and social impacts (Line 50).

I would suggest that the paper focus on the restaurant business vs, the vague term “social impact.”

I do not understand categorizing NPIs based on need and the terms used – mild, focused, strong.

I keep thinking a case study approach would be good for this paper – tell the story of what you found when you examined the Tokyo experience in the first 6 months of the pandemic.

I think the paper would be excellent if you introduce each type of data and explain what you found and then use causal loop diagramming as a qualitative analytical lens.

Line 186-218: this section lacked focus

I do not know why the sub-headings contain the words causal loop diagram.

Line 221 – what is a system review?

Hotels, restaurants, economic sector, schools, commuters, medical treatments, sales data   – I just couldn’t follow all the lines of thinking.

People flow? this was not addressed in the Intro or Methods.

Without a substantive discussion of data sources like eWOM , it is hard to discern quality?

Line 280-309: Again, how were the feedback loops developed? What was the process? The explanation of the CLD feedback loops were very hard to understand.

Figure 3. I am very familiar with CLDs and I could not understand how some feedback loops were created and labeled. In my opinion there must be a direct link to the data (not vague explanations as above) and then an explicit discussion of most of the variables.

Section 3.7 was inadequate in terms of the clear links to the data, the analysis process and how you conducted quantitative modelling. I cannot read Figure 4 with any clarity.

Section 3.8 Here you “posit” and as in my comments above, I wondered whether a clear set of hypotheses in the introduction would help organize the paper for readability?

Discussion

I am not sure that “fear of economic stagnation” was clearly illustrated in the paper. Lack of clear links to all the previous sections. Not sure how reference to pre-existing immunity helps this paper. In the end I was so confused as to what your data showed and what your assumptions were that I could not see the value of this research. I love the idea of examining the key factors that enabled or hindered COVID-19 transmission in conjunction with what NPIs were implemented and what happened to restaurants in Tokyo.

Author Response

July 17, 2021

Dear Reviewer,

First of all, we’d like to express to you our gratitude. Your comments and suggestions are invaluable for us to improve our paper.

We believe that we did our best in clarification and improvements according to all of your comments and suggestions as below, and we hope that our responses meet your expectations and your intentions.

We highly appreciate your cooperation.

Warm Regards,

Kota Kodama, PhD

Graduate School of Technology Management, Ritsumeikan University

2-150, Iwakura-cho, Ibaraki, Osaka, 567-8570, Japan

+81-72-665-2448

[email protected]

ID

Comments and Suggestions

Response

Reviewer

1-1

Introduction

I found this section to be weak because it did not lay our clearly what the research questions were to be addressed in the study.

We re-organized introduction, which is consisted of (1) aspect of choosing region of interest (Japan) (lines 47-61), (2) aspect of choosing industry (restaurant industry) and typical metrics (eWOM) (lines 67-84) (3) aspect of dealing with complexity (line 89-104).

Reviewer

1-2

Specific areas that were unclear:

Line 55-59: this long sentence contains far too many ideas.

This long sentence is taken apart into (1) importance of investigate into the potential reason of low numbers (line 54-56), (2) first hypothesis to be addressed in this research (line 56-57), and (3) future importance of this investigation (line 57-61).

Reviewer

1-3

Line 65-80: the mix of ideas are difficult to understand what the key point is.

This paragraph was re-organized into 3 paragraphs, (1) restaurant industry (line 67-70), (2) supportive information in other service industry (line 71-76) and (3) implication of eWOM on sales (line 77-84).

Reviewer

1-4

Line 83: what is sustainable infection control? Definitions are needed throughout the paper.

Sentence was revised to “control infections with minimized negative economic impact on restaurant industry” for clarity (line87-88).

Reviewer

1-5

Line 89-98: this information should be in Materials and Methods.

A part of this paragraph (information on used data) is moved to Materials and Methods section, and taken into appropriate position in section 2.2 (line 135-138).
This paragraph was re-organized to clarify research strategy of using system dynamics for clarity (line 95-104).

Reviewer

1-6

Materials and methods

Line 116-122: How were the causal loop diagrams developed? What was the exact process? No idea what bottom-up means. How was data summarized? How was the stock and flow diagram created? What was the data analysis process?

Description on development of causal loop diagram is added in section 2.2 (line 125-132). It was clarified that causal loop was used to clarify the possible relationships between Japanese NPIs and items of interest based on preceding studies.

Reviewer

1-7

Line 137: what economic effect model?

This was re-written to clarify that this is a subpart of stock-flow model (line 153).

Reviewer

1-8

Line 139: assumed? Could you revise the paper and clearly articulate hypothesis examined?

Manuscript was revised to include research hypothesis in introduction (line 56-57, 80-83). Term “assumed” originally written meant the assumption put in the model (based on the concept of “mind model” of system dynamics) and this assumption was derived from the idea that visiting to restaurants with eWOM is considered to be an activity with strong intention and is not dependent on people flow and lowered people flow by remote work. This clarification was added (line 155 – 158).

Reviewer

1-9

Line 145: real and simulated data? Not idea what this means here. The paper needs a clear section on how each data set was identified and analyzed.

The purpose of this comparison was to check the reality of model structure, and this explanation was added (line 165-168).

Clarification in analytical strategy was thoroughly re-organized with additional information in section 2.2 and 2.3. Description on simulation was separated into section 2.4.For quantitative stock-flow model, parametrization is provided as Supplementary Table S1.

Reviewer

1-10

Results

Line 170: I don’t think there is any debate anymore about aerosol transmission!

We agree to this point of view and this information was deleted.

Reviewer

1-11

This first paragraph does not clearly explain results.

The intention was to explain the process of construction of causal loop diagram by reviewing preceding studies. This paragraph was re-constructed for clarity (line 260-268).

Reviewer

1-12

I had a very hard time reading the results section and often myself questioning how various paragraphs fit with what I understood to be the study design – comparing “dynamics of infection, NPIs, and social impacts (Line 50).

The manuscript was thoroughly re-constructed to clarify research structure (observation (section 3.1), causal loop based on preceding studies and observation if applicable (sections 3.2 through 3.7) quantitative model construction (sections 3.8 and 3.9), hypothetical modeling & simulation (section 3.10)).

Reviewer

1-13

I would suggest that the paper focus on the restaurant business vs, the vague term “social impact.”

We changed the term “social impact” or related expression to “restaurant industry / business” (line 86 and 88) and checked the whole manuscript for consistency.

Reviewer

1-14

I do not understand categorizing NPIs based on need and the terms used – mild, focused, strong.

The original intention was that NPIs can be categorized on the basis of restriction strength, and Japan initially employed tiered approach from weak to strong. The sentence was revised for clarity (line 276-281).

Reviewer

1-15

I keep thinking a case study approach would be good for this paper – tell the story of what you found when you examined the Tokyo experience in the first 6 months of the pandemic.

Based in this valuable suggestion, observational finding is placed in the first part of the results section (section 3.1).

Reviewer

1-16

I think the paper would be excellent if you introduce each type of data and explain what you found and then use causal loop diagramming as a qualitative analytical lens.

Observational finding is placed in the first part of the results section. Causal loop was used as method for exploring causes and potential feedbacks (in classic system dynamics context) using preceding studies and observations. Although the idea to use causal loop analysis as quantitative analytical lens is attractive, this study employed mind model concept and this was clarified in line 129.

Reviewer

1-17

Line 186-218: this section lacked focus

This section was intended to show outline of Japanese NPIs. The order of paragraphs were put in a time series (line 282-297).

Reviewer

1-18

I do not know why the sub-headings contain the words causal loop diagram.

These section (sections 3.2 through 3.7) describes the construction of causal loops mainly based on preceding studies. Explanation saying the section deals with causal loop was added in the first part of each section.

Reviewer

1-19

Line 221 – what is a system review?

This should have been systematic review. This sentence was corrected (line 317).

Reviewer

1-20

Hotels, restaurants, economic sector, schools, commuters, medical treatments, sales data   – I just couldn’t follow all the lines of thinking.

The focus is on NPIs and restaurant industry. Intermediator was people flow. The whole manuscript is re-constructed to clarify the lines of thinking.

Reviewer

1-21

People flow? this was not addressed in the Intro or Methods.

People flow is described in methods section (section 2.1, line 116-117).

Reviewer

1-22

Without a substantive discussion of data sources like eWOM , it is hard to discern quality?

Feature of e-WOM, reported by qualified reporters, was added in section 2.1 (line 112-115).

Reviewer

1-23

Line 280-309: Again, how were the feedback loops developed? What was the process? The explanation of the CLD feedback loops were very hard to understand.

Feedback loop was developed based on preceding studies and observation of NPIs. Sections 3.2 – 3.6 were reconstructed to clarify this process.

Reviewer

1-24

Figure 3. I am very familiar with CLDs and I could not understand how some feedback loops were created and labeled. In my opinion there must be a direct link to the data (not vague explanations as above) and then an explicit discussion of most of the variables.

Causal loop was used as method for exploring causes and potential feedbacks (in classic system dynamics context) using preceding studies and observations. Solid arrows indicate relationships supported by policies, theories, or previous studies, while broken lines indicate hypothetical relationships reflecting mind model in system dynamics context. Caption of Figure 3 was amended for clarity.

Reviewer

1-25

Section 3.7 was inadequate in terms of the clear links to the data, the analysis process and how you conducted quantitative modelling. I cannot read Figure 4 with any clarity.

(Now this section is 3.8) Analytical process was clarified in sections 2.3 and 2.4. Structure of the section  was re-constructed for clarity (line 334-339).

Reviewer

1-26

Section 3.8 Here you “posit” and as in my comments above, I wondered whether a clear set of hypotheses in the introduction would help organize the paper for readability?

(Now this section is 3.9) This was an assumption put to build models.  Word “posit” was inadequate and replaced with the phrase “was built to explain” (line 419, 427, 435).

Reviewer

1-27

Discussion

I am not sure that “fear of economic stagnation” was clearly illustrated in the paper. Lack of clear links to all the previous sections. Not sure how reference to pre-existing immunity helps this paper. In the end I was so confused as to what your data showed and what your assumptions were that I could not see the value of this research. I love the idea of examining the key factors that enabled or hindered COVID-19 transmission in conjunction with what NPIs were implemented and what happened to restaurants in Tokyo.

The term economic stagnation was substituted by the information that the economic sector may pressure the government to withdraw stronger NPI measures that can negatively impact economies, which appeared in section 3.3.

Whole manuscript was re-constructed to clarify research hypothesis, preceding studies, real data, model construction (including model assumption), output from the model, and simulation using the model.

Reviewer 2 Report

In this study, the elaboration of theoretical transmission models provides useful indications for the selection of efficient strategies for the control of SARS-CoV-2 infection. Predictive models are based on plausible assessments. The authors demonstrate that strong short-term interventions were efficient ways of controlling the spread of disease over short periods of time with minimum economic effects, but early lockdown terminations will still reboot the viral spread and continuous intervention, and continued attention (and probably massive vaccination) are necessary, as evidenced by the evolution of the Japanese situation, where just today it was necessary to take new strong restrictive measures

Author Response

July xx, 2021

Dear Reviewer,

First of all, we’d like to express to you our gratitude. Your comments and suggestions are invaluable for us to improve our paper.

We revised our manuscript according to suggestions from all reviewers. We believe that we did our best in improving our manuscript, and we hope that our current version meet your expectations.

We highly appreciate your cooperation.

Warm Regards,

Kota Kodama, PhD

Graduate School of Technology Management, Ritsumeikan University

2-150, Iwakura-cho, Ibaraki, Osaka, 567-8570, Japan

+81-72-665-2448

[email protected]

ID

Comments and Suggestions

Response

Reviewer

2-1

In this study, the elaboration of theoretical transmission models provides useful indications for the selection of efficient strategies for the control of SARS-CoV-2 infection. Predictive models are based on plausible assessments. The authors demonstrate that strong short-term interventions were efficient ways of controlling the spread of disease over short periods of time with minimum economic effects, but early lockdown terminations will still reboot the viral spread and continuous intervention, and continued attention (and probably massive vaccination) are necessary, as evidenced by the evolution of the Japanese situation, where just today it was necessary to take new strong restrictive measures.

Thank you for reviewing and comments. We further refined the manuscript based on the suggestions of all reviewers.